# Adaptive Propagation Graph Convolutional Networks Based on Attention Mechanism

**Chenfang Zhang** [1],[*] , **Yong Gan** [2] **and Ruisen Yang** [1]

1   School of Computer Communication and Engineering, Zhengzhou University of Light Industry, Zhengzhou 450002, China
2   School of Computer Communication and Engineering, Zhengzhou Institute of Engineering and Technology, Zhengzhou 450002, China
*   Correspondence: 8459055287wudi@gmail.com

**Abstract:** The main steps in a graph neural network are message propagation and aggregation between nodes. Message propagation allows messages from distant nodes in the graph to be transmitted to the central node, while feature aggregation allows the central node to obtain messages regarding its neighbors and update itself, so that it can express deep-layer features. Because the graph structure data have no local translation invariance, the number of neighbors of each central node is different, and there is no order, there are two difficulties: (1) how to design a reliable message propagation method to better express all network topologies; (2) how to design a feature aggregation function so that it can weigh local features and global features. In this paper, a new adaptive propagation graph convolutional network model based on the attention mechanism (APAT-GCN) is proposed, which enables GNNs to adaptively complete the process of message propagation and feature aggregation, according to the neighbors of the central node, and set the influence degree of local and global messages on the aggregation of the central node. Compared with other classical models, this method is superior to the baseline model and can improve the accuracy of node- and graph-level classification tasks in downstream tasks.

**Keywords:** graph neural networks; convolutional network; attention mechanism; adaptive propagation





## 1. Introduction

In the last decade, convolutional neural networks have made significant achievements in images, videos, and texts because they can extract potentially effective information from Euclidean spatial data [1]. The reason is that convolutional neural networks can extract potentially useful information from Euclidean spatial data. In the case of images, for example, which are regular lattices with structural information, convolutional neural networks can exploit their translational invariance and local connectivity, so that a particular type of convolutional kernel can always extract meaningful features from image [2].

However, in addition to structured data, such as images, video, and text, there is also a large amount of unstructured data in real life, such as social networks, citation networks, protein networks, and traffic networks with graph structures. In these non-Euclidean spaces, there are tasks such as node classification [3], connection prediction [4], graph classification [5], and others. For example, in e-commerce, graph-based learning systems can use the interaction between users and products to make highly accurate recommendations; in chemistry, molecules are modeled as graphs whose biological activity needs to be identified for drug discovery; in citation networks, papers are interlinked by citations and need to be classified into different groups. For these irregular data objects, ordinary convolutional networks are not as effective as they could be. Since graph structure data may be irregular, a graph may have unordered nodes of unequal size, and nodes from a graph may have different numbers of neighbors. This leads to some important operations

(e.g., convolution) that are easy to compute in the image domain, but difficult to apply to graphs [6].

## 2. Related Work

Researchers have started to migrate classical CNN networks to non-Euclidean spaces, and the development of graph convolutional neural networks has been divided into two main directions: spectral domain- and null domain-based [7].

Based on the spectral domain of the graph convolutional neural network using graph signal theory, through the Fourier transform to convert the graph structure data into the spectral domain, using the Laplace matrix and other operators to carry out a process of noise reduction, the processing is completed and then converted to the original empty domain through the Fourier transform [8]. The key in this method is filter selection, and different choices lead to different results, resulting in different GCN algorithms, based on the spectral domain [9].

The first is the spectral CNN proposed by Le Cun in 2014, which mimics the properties of CNNs by overlaying multilayer networks, adding nonlinear activation functions, and defining graph convolution kernels at each layer for forming a graph convolutional neural network, with all operations performed in the spectral domain of the nodes [10]. However, its explicit use of Laplacian matrix eigen decomposition and dense eigenvector matrix multiplication is difficult to ground on large-scale graphs, thus leading to the problem of non-localization of the convolution kernels. The Chebyshev spectral CNN proposed in 2016 parameterizes the convolution kernel by K Chebyshev polynomial approximations [11], and the new convolution kernel is a polynomial combination of the eigenvalues of the original Laplacian matrix, so that local information is taken into account. So that only the parameters and polynomials to be learned are retained in the convolution theorem results, the number of parameters and computational complexity is greatly reduced, and graphical convolutional neural networks become practical. However, since GNNs based on spectral domains essentially make use of the Laplace matrix of a particular graph, GNN models learned on a particular graph cannot be used for other graphs and new nodes. Wavelet neural networks (GWNNs) were proposed in 2019 [12]. The wavelet transform was used, instead of the Fourier transform, to implement the convolution theorem to transform and extract features from the graph signal, whose basis low can be obtained by Chebyshev polynomial approximation, thus avoiding the high cost of the characteristic decomposition of the Prussian matrix, and with localization, making the wavelet transform matrix very sparse and greatly reducing the computational effort, which can be flexibly applied to different task scenarios by adjusting the hyperparameters in the heat kernel function.

Based on the fact that null domain of the graph convolutional neural network did not start from the graph signal theory, but directly from the nodes in the graph considered, by defining the aggregation function to aggregate the features of each central node and its neighboring nodes, they are generally divided into two steps: the aggregation function will act on each node and its neighboring nodes to get the local structure of the node expression; the update function will act on itself and the local structure of the expression to get the current new expression of the node [13]. The update function is applied to itself and the local structure expression to obtain a new expression for the current node. When dealing with image problems, convolutional neural networks use a fixed size kernel to extract features from images. When faced with graph structure data, it is not feasible to still use a fixed-size kernel as the number of first-order neighbors of each node in the network is different, because the perceptual field of the convolutional kernel will be different and nodes of the graph lack order, so a more conventional approach is to select the center node and a fixed number of neighboring nodes, thus sorting the selected nodes. From another perspective, spatially-based ConvGNNs share the same philosophy as RecGNNs, in terms of message propagation, i.e., the convolutional operation of a spatial graph is essentially the propagation of node information along the edges.

In the PATCHY-SAN, proposed in 2016 [14], for each input graph, the PATCHY-SAN method first determines a sequence of nodes. Then, for each node in the sequence, a neighborhood of exactly k nodes is extracted and normalized, and the normalized neighborhood is used as the perceptual field of the current node; finally, similar to the perceptual field of a CNN, some feature learning components (e.g., convolutional layers, sense layers) can be applied to the normalized neighborhood. PATCHY-SAN is computationally efficient, natively supports parallel computation, and can be used for large graphs. Graph-SAGE [15], proposed in 2017, does a separate linear transformation of the node's attribute features and sampled neighboring node features and then merges the two and performs another linear transformation to obtain the target node's feature representation. Finally, the resulting target node representation can be used for downstream tasks. The introduction of a fixed number of random neighborhood samples can limit the number of nodes to be processed to a certain interval, thus eliminating the need to input the entire graph, improving computational efficiency, and transforming the straightforward node representation of only one local structure into a node-inductive representation corresponding to multiple local structures, effectively preventing training overfitting and enhancing generalization capabilities.

## 3. Models and Definitions

### 3.1. APAT-GCN Model

The core of the method introduced in Chapter 2 is to aggregate the surrounding information to the central node; then, after multiple layers of convolution, the central node can obtain the message of the distant nodes. However, after multiple convolutions, the features aggregated by all the central nodes become increasingly smooth, i.e., the features are similar across nodes, which makes the subsequent downstream tasks not perform well. The reason for this is that the above graph convolutional neural network method cannot obtain global information and can only stay on local aggregation. Moreover, the above methods do not take the different effects of different neighboring nodes on the central node when aggregating into account, but simply add them up, which will result in the central node not capturing the information that is useful to it.

To address these issues, a new graph convolutional neural network model is proposed in this paper: adaptive propagation graph convolutional network, based on attention mechanism APAT-GCN. This model has an adaptive propagation stopper at each node that can individually calculate whether the node needs to continue with the next layer of convolution, so that each node performs a different number of layers of convolution, as we will argue in the experimental section. The need to assign different layers of convolution to each node is demonstrated in the experimental section. In addition, the model makes use of an attention mechanism to assign different weights to each node, so that the central node can find the information that is useful to it when convolving. To alleviate the transition smoothing problem during deep convolution, a fraction of unqualified nodes is actively discarded as disconnected in each neighbor node sampling phase.

The model first samples the neighbors of the central node, adds the obtained neighboring nodes to the attention coefficients, and then assigns different weights to them during the aggregation process. After the aggregation is completed, the adaptive propagation stopper of the central node determines whether the next convolution operation is required; if not, the final feature representation of the node is derived. In Figure 1, $h_v$ and $h_{N_i}$ represent the features of the central node's neighbors after the aggregation function, and the dashed lines indicate the node connections that were not sampled. More details are described in Figure 2. The adaptive propagation graph convolutional network, based on the attention mechanism proposed in this paper, has the following three contributions:

1. Setting a different number of convolutional layers for each node, which can speed up training, while reducing memory consumption.
2. Sampling of neighboring nodes and discarding some of them can alleviate the problem of over-smoothing during deep convolution.

3.　　Introducing an attention mechanism, so that the central nodes can access more information useful to them when aggregating.

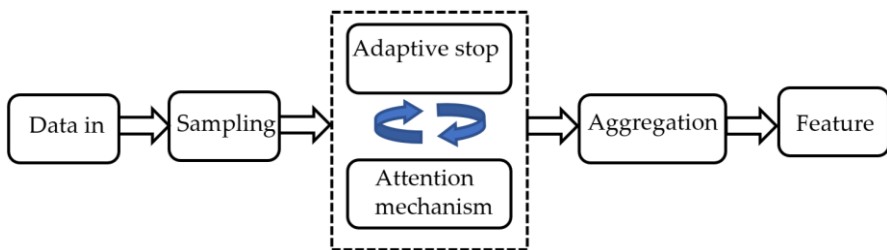

**Figure 1.** APAT-GCN framework.

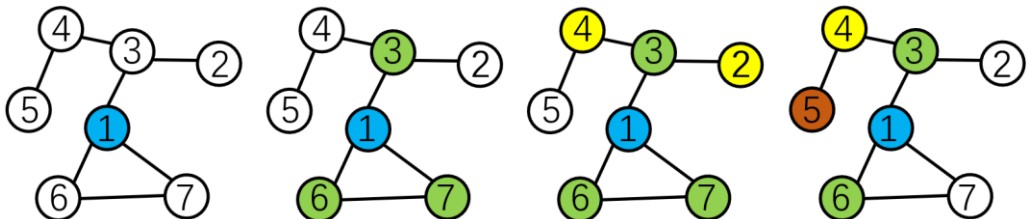

**Figure 2.** The picture shows the propagation process of a node. In the first step, node 1 propagates one step and finds 3, 6, and 7. In the second step, it propagates two steps and finds 4 and 2. In the third step, it propagates three steps. Then, 5 is found, but, at this time, the adaptive stop propagator of nodes 2 and 7 stops the propagation of nodes 2 and 7, so the features of these two nodes should be removed in this step.

### 3.2. Definition of Graph

A graph is a collection of entity relationships, e.g., an individual in a social network is a node and relationships with other people are edges, each article in a citation network is a node, and the citation relationships between articles are edges. The graph is, thus, defined as $G = (V, E)$, where $V$ is the set of all nodes in the graph $V = \{v_1, v_2, \ldots, v_n\}$, $n$ is the number of nodes in the graph $G$, and $E$ is the set of edges in the graph $G$ $E = \{(v_i, v_j) | v_i, v_j \in V\}$, with the edges representing the relationship between two entity objects. We use $X$ to denote the features of nodes on graph $G$, where $X \in R^{n \times d}$ denotes the features of the $i$th node. The adjacency matrix $A_{ij} = \begin{cases} 1, (v_i, v_j) \in E \\ 0, \text{ otherwise.} \end{cases}$ and degree matrix $D_{ii} = \sum_j A_{ij}$ can be defined based on the connections between the nodes in the graph. For the node classification task, we know some of the nodes $T \in V$ and have labeled the nodes $i \in T$ with their true values $y_i$ to classify the other unlabeled nodes. The same is true for the graph classification task.

### 3.3. Designing and Training Deep Graph Convolutions

Message propagation neural network MPNN [16], a generic form of GCN, for node-specific updates, is summarized as follows.

$$h_i = F(\{f(x_j) | j \in N_i\}) \tag{1}$$

where $f(x_j)$ is the node message propagation method, and $F$ is the node aggregation method. The method mentioned in the previous section is a specific design improvement of $f(x_j)$ and $F$. Once the specific network has been designed, the cross-entropy loss can be used to optimize our model parameters, as with other convolutional neural networks.

$$f^* = \text{argmin}\left\{\sum_{i \in T} y_i \cdot \log(f(\mathbf{x}_i))\right\} \tag{2}$$

## 4. Adaptive Aggregated Graph Convolutional Network

### 4.1. Graph Centrality Sampling

Before the graph convolutional neural network can start working, the nodes used need to be sampled. In graph-SGAE, the authors propose a small batch training method using random sampling, but the results of random sampling may not be the most valuable nodes needed for the model. To make the sampling as fair and convincing as possible, this paper uses an alternative sampling method, using the graph centrality [17] as a criterion for evaluating the importance of a sample. Graph centrality is the inverse of the maximum value of the shortest path length between a point and other points in the connectivity component of the point. The result of this calculation indicates whether the point is at the center of the graph or not. In the experimental section, we compare the effects of graph centrality sampling and random sampling of the model results.

For graph structure data, the importance of the nodes can be calculated using the defined adjacency matrix and then ranked, from which, the important set of nodes can be selected and used for training. As shown in Figure 3, the sampling process is as follows.

1. Adjacency matrix of the input graph structure data.
2. Calculate the centrality of each node in the adjacency matrix, and then obtain a portion of the nodes with higher scores from their neighbors.
3. Remove the obtained nodes from the adjacency matrix to obtain a new adjacency matrix.
4. Repeat steps 2 and 3, until the number of neighboring nodes obtained is sufficient.
5. Feed all selected nodes into the model for training.

$$C_G(v_i) = \frac{1}{\max d(v_i)} \tag{3}$$

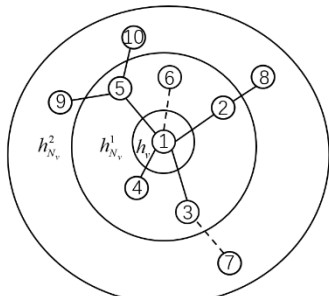

**Figure 3.** Neighborhood sampling.

### 4.2. Adaptive Propagation Stop

In general, graphical convolutional neural networks have a single constant maximum number of propagation steps set and shared by each node, whereas our model binds an adaptive propagation stopper to each node, as shown in the Figure 4, which can be changed during training to indicate whether the node needs to continue propagation and can be made to adjust the propagation range by setting hyperparameters to achieve both local and global propagation. The stopping probability of each node can be expressed as follows.

$$h_i^k = S(Qh_i^k + q) \tag{4}$$

where $Q$ and $q$ are the parameters used for training and represent the probability that a node $i$ will stop at the $k$ layer. To ensure that the number of propagation steps remains reasonable, we restrict the range of propagation. First, we fix a maximum number of propagation steps T. Second, we define the budget of the propagation process using the sum of the number of stopping propagations.

$$K_i = \min\left\{ k' : \sum_{k=1}^{k'} h_i^k >= 1 - \varepsilon \right\} \tag{5}$$

where $1 - \varepsilon$ is used to ensure that propagation can be achieved at least once and stops when the propagation step $k = K_i$ is reached. Thus, the final expression for the stopping probability after combining the various scenarios is as follows.

$$p_i^k = \begin{cases} R_i = 1 - \sum\limits_{k=1}^{k_i-1} h_i^k, if\ k = K_i\ or\ k = T \\ \sum\limits_{k=1}^{K_i} h_i^k, otherwise. \end{cases} \tag{6}$$

this calculates the stopping probability of each node, which is then used to choose whether to stop or continue propagation when node propagation is performed; when propagation is stopped, the features of that node are no longer updated.

$$h_i^{k+1} = \frac{1}{K_i} \sum\limits_{k=1}^{K_i} p_i^k h_i^k + (1 - p_i^k) h_i^{k-1} \tag{7}$$

where $h_i^{k+1}$ denotes the node characteristics of node $i$ after the $k + 1st$ propagation, after adding the adaptive propagation stopper. It can be seen that, if the node is judged to have stopped after the kth propagation, then the result of the kth update is utilized for subsequent updates.

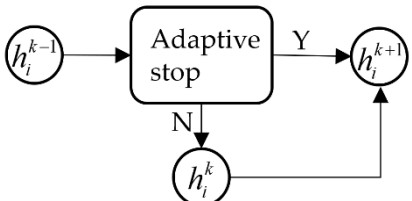

**Figure 4.** Adaptive propagation stop unit.

The adaptive propagation stopper we designed takes its cue from the RNN model [18], inspired by the fact that node updates at each layer of the network need to make use of the features of the nodes updated in the previous layer.

$$\begin{aligned} h_i^o &= x_i \\ h_i^1 &= f(h_j^0, W^o | j \in N_i) \\ h_i^2 &= f(h_j^1, W^1 | j \in N_i) \\ &\dots \end{aligned} \tag{8}$$

to allow the central node to freely receive both global and local information, we set the propagation cost $S_i$, which indicates the number of propagation steps required for the update of the $i$th node.

$$S_i = K_i + R_i \tag{9}$$

if $L$ is used to indicate the cost of loss, then use:

$$\hat{L} = L + a \sum\limits_{i \in V} S_i \tag{10}$$

$$S_i = K_i + R_i \tag{11}$$

where $a$ represents a balancing factor over local and global propagation, i.e., the extent of propagation that can be controlled by the adjustment of $a$.

### 4.3. Attention Mechanism

In order for the central node to obtain really useful information, we need to calculate the influence of the neighboring nodes of the central node $v_i$ on itself (including $v_i$ itself), assign different weights to the neighboring nodes, and, for the central node $i$, this is shown in Figure 5, calculate the correlation coefficient between the sampled neighboring nodes and it.

$$e_{ij} = \varphi(Wh_i \| Wh_j) \tag{12}$$

where $j \in N_i$. The correlation coefficient is then normalized to obtain the attention coefficient.

$$\alpha_{ij} = \frac{\exp(LeakyRelu\ e_{ij})}{\sum\limits_{k \in N_i} \exp(LeakyRelu\ e_{ik})} \tag{13}$$

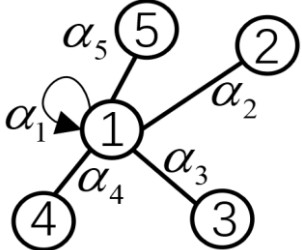

**Figure 5.** Schematic diagram of the attention mechanism.

Once the attention coefficients are obtained, they are put into the aggregation function, together with the node features and pre-training weight matrix, and then the feature fusion operation can be performed.

$$h_i^k = \sum_{j}^{|N_i|} \alpha_{ij}^k h_{N_i} W^k \tag{14}$$

where $\alpha_{ij}^k$ denotes the attention coefficient of neighbor node $j$ to central node $i$ at the $k$ th layer of the convolutional network, and $W^k$ denotes the weight matrix of central node $i$ to each neighbor node at the $k$ th layer of the convolutional network. A multi-headed attention mechanism can also be introduced to enhance the expression of feature fusion [19].

## 5. Experimental Analysis

### 5.1. Data Set and Experimental Setup

There are a large number of real-life practical tasks that can be abstracted into graph-structured data, and the application scenarios are numerous. Graphs contain nodes and edges. The downstream tasks of graph neural networks are mainly at the node, edge, and graph levels. At the node level, node-specific classification can be performed, for example, in citation datasets and to classify similar papers. At the edge level, link prediction can be performed, for example, to infer whether two people know each other in social networks. At the graph level, graph classification can be performed, for example, to classify chemical formulae of the same family into one class [20]. For the different processing tasks, the datasets used in the experiments in this paper are shown in Table 1.

**Table 1.** Data sets.

| Type of Task | Data Sets | Number of Nodes | Number of Sides |
|---|---|---|---|
| Node classification | Core | 2708 | 5429 |
| | CiteSeer | 3312 | 4732 |
| Figure classification | Protein | 43,471 | 162,088 |
| | Reddit-5K | 122,737 | 265,506 |

The Cora dataset consists of 2708 papers related to machine learning, each of which cites, or is cited by, at least one other paper in the dataset, grouped into seven categories. The CiteSeer dataset contains 3312 scientific publications, grouped into six categories. The citation network consists of 4732 links. Each publication in the dataset is described by a word vector with a value of 0 or 1, which indicates the presence or absence of the corresponding word in the dictionary. Each node in the Protein dataset is an element of a secondary structure, and an edge exists if two nodes are adjacent nodes in the amino acid sequence or 3D space. Each node in the Reddit-5K dataset represents a user, and each graph represents a post, where an edge exists if a user replies to another user's comment. The Reddit-5K used in this paper has 5000 graphs that are divided into two graph categories.

### 5.2. Over-Smoothing Problems

The graph node representation obtained by the previous algorithm shows that there is a transition smoothing problem because the central node introduces too much noise during aggregation, and the long-distance node features will also be propagated to the central node, resulting in the characteristic representation obtained by the final aggregation of the central node, which is very similar and not easy to distinguish; in this process, it is difficult to avoid the introduction of long-distance node information that is not related to the central node feature representation during aggregation, and the existence of these problems seriously affects the completion of the downstream task of the graph.

The Figure 6 shows some of the over-smoothing problems that exist in graph convolutional neural network models. In this paper, we test the MAD and F1 indices under the number of iterations of different graph neural network layers in graph-SAGE, GAT, and APAT-GCN. Among them, the MAD metric measures the average distance between node representations, while the F1 metric measures how accurate the model is for node classification issues.

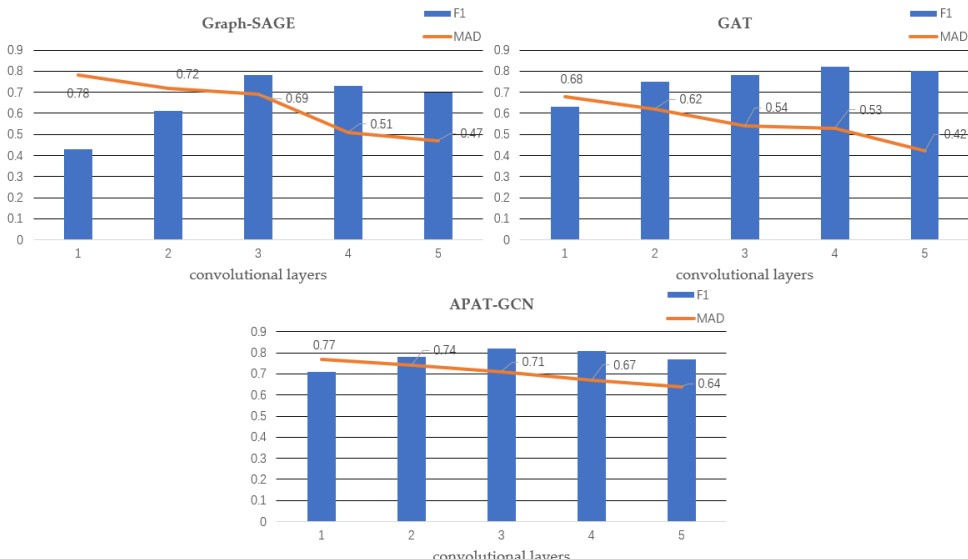

**Figure 6.** The relationship between the MAD and F1 indicators of the three models and number of layers.

It can be seen from the results that in the graph-SAGE and GAT models, when the number of neural network layers increases, the node feature representation becomes more and more similar, and the model effect begins to decline. This result verifies that the effect of the existing graph neural network model degrades, due to over-smoothing problems, when the number of layers increases. In APAT-GCN, the oversized problem is mitigated, suggesting that the model's sampling and aggregation methods are, indeed, effective.

*5.3. Parameter Settings for the Baseline Model*

In this section, we will set up the hyperparameters of three other models used for the comparison experiment: graph-SAGE, AP-GCN, and GAT to achieve the best results in the node classification task. The results are shown in Figure 7.

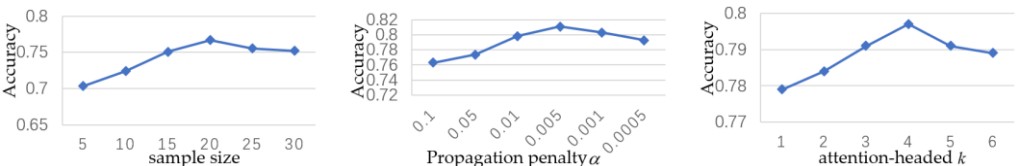

**Figure 7.** The accuracy of node classification in the core dataset after adjusting the hyperparameters of graph-SAGE, AP-GCN, and GAT.

The graph-SAGE model uses $k$ to confirm the propagation range and $s_i$ to determine the number of samples of the ith-order neighbors. We use the configuration of the original text, $k = 2$, $s_1 = s_2$, so that the center node can get the information of the second-order neighbor at most. The AP-GCN model uses the propagation penalty parameter $\alpha$ to dynamically adjust the range of propagation and sets the maximum number of propagation steps to 10. GAT mainly adjusts the degree of aggregation of node information by setting the attention head, but the attention head is too large. It is not only slower in training, but also causes overfitting.

*5.4. Comparison of Experimental Effects*

In this subsection, we use graph-SAGE, AP-GCN, GAT, and the adaptive algorithm model proposed in this paper to do benchmarking tasks on four datasets, Core, CiteSeer, Protein, and Reddit-5K, respectively. Among them, Core and CiteSeer are node classification tasks that classify articles into different types. Protein is a graph classification task that classifies protein molecules into different types, based on their roles, and Reddit-5K is a task that classifies different communities, based on the topics of posts [21], the results of which are shown in Table 2.

**Table 2.** Prediction results for the four data sets (micro-averaged F1).

| Models | Core | CiteSeer | Protein | Reddit-5K |
|---|---|---|---|---|
| Graph-SAGE | 0.778 | 0.791 | 0.921 | 0.908 |
| AP-GCN | 0.806 | 0.811 | 0.917 | 0.897 |
| GAT | 0.802 | 0.801 | 0.933 | 0.907 |
| APAT-GCN [1] | 0.811 | 0.816 | 0.941 | 0.912 |

[1] Means the model we propose.

The experimental results, expressed in terms of micro-averaged F1 values, show that the adaptive algorithm shows relatively good results on all four datasets. The reason for this is that the algorithm combines an adaptive stopping unit and attention mechanism that is able to globally notice the information that has the greatest impact on the central node and produces less noise.

To demonstrate that the graph centrality sampling results are better for training than other sampling methods, we did a comparison experiment at Core. The results are shown in Table 3.

**Table 3.** Comparison of sampling results (micro-averaged F1).

| Sampling Method | Micro-Averaged F1 (Core) |
|---|---|
| Random | 0.304 |
| Deep walk | 0.742 |
| Chart center degree sampling | 0.799 |

The results show that the graph centrality sampling approach can indeed improve the accuracy of the model on classification problems.

*5.5. Adjustment of Hyperparameters*

To enable the model to obtain a better result, in terms of global and local information acquisition, we set the hyperparameter *a* in Equation (10), which can be adjusted so that the perceptual field of information transfer can be expanded or reduced. This hyperparameter was adjusted on the node classification problem using the Core dataset. The result of adjusting the hyperparameters is shown in Figure 8.

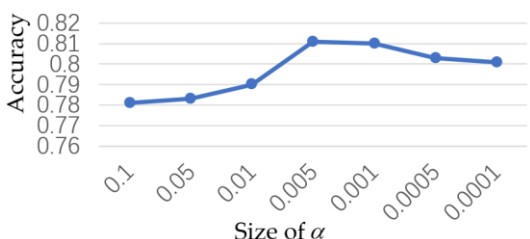

**Figure 8.** Adjustment of hyperparameter *a*.

It can be seen that at $a = 0.005$, the model captures the best global information, without causing over-smoothing problems.

## 6. Conclusions

In this paper, we propose a new graph convolutional neural network model, APAT-GCN, which uses graph centrality sampling in the sampling phase, an adaptive propagation stopper in the message propagation phase, and an attention mechanism at the feature aggregation nodes, according to three important steps of graph convolutional neural networks, respectively, thus making APAT-GCN better able to handle large data volumes of graphs, in order to mine the nodes in the graph. This allows APAT-GCN to better handle large data volumes, mine deeper information about the nodes in the graph, and aggregate global information.

The model can be easily applied to tasks that process other unstructured data, such as commodity recommendation systems, social networks, and public opinion evolution monitoring. At present, the research on dynamic graph structure is not yet mature enough. If GCN can be successfully applied to the dynamic graph structure, it is believed that this will make the application field of GCN more extensive [22]. Our future work will focus on making the GCN model available to run on dynamic graphs.

**Author Contributions:** Conceptualization, Y.G.; methodology, Y.G.; software, C.Z.; validation, C.Z. and R.Y.; resources, C.Z.; writing—original draft preparation, C.Z.; writing—review and editing, C.Z. All authors have read and agreed to the published version of the manuscript.

**Funding:** This research was funded by the Nation Nature Science Foundation of China (NSFC), (Nos. 61572445, U1804263).

**Institutional Review Board Statement:** Not applicable.

**Informed Consent Statement:** Not applicable.

**Data Availability Statement:** Not applicable.

**Conflicts of Interest:** The authors declare no conflict of interest.

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
