# Peer review of "Adaptive Propagation Graph Convolutional Networks Based on Attention Mechanism"

_information, doi:10.3390/info13100471_

Round 1
Reviewer 1 Report
1. The paper presents a new adaptive prop- 15 agation graph convolutional network model based on the attention mechanism (APAT-GCN), which is a innovative topic of interest to the researchers in the related areas.
2. The paper have noticed two difficulties, how to design a reliable message propa- 13 gation method to better express all network topologies and how to design a feature aggregation 14 function so that it can weigh local features and global features.
3. The method of domain adaptation proposed by this paper works very well for enabling GNNs to adaptively complete the process of message propagation and feature aggregation according to the neighbors of the central node. Compared with other classical models, this method is superior to the baseline model, and can improve the accuracy of node-level and graph-level classification tasks in downstream tasks.
4. The structure of the paper is clear, and charts and formulas are used to explain the method proposed in this paper. At the same time, it is compared with the previous training methods, which shows that this method is superior in performance, and pictures can be added appropriately to explain the model.
5. The title is accurate, concise and closely related to the theme. The summary part includes the purpose, methods, results and conclusions of the research, and the keyword selection is appropriate. The introduction clearly reflects the research background, research progress in this field, remaining deficiencies or unresolved problems and what problems the author intends to solve and how to solve them.
6. The selection of data sets is full and true, and the corresponding analysis and demonstration are rigorous. The results obtained are credible, and the analysis and comparison of the results are sufficient. Draw a general conclusion according to the research results, which has application value. From the research process, the conclusion is persuasive.
7. The paper lacks an objective analysis of the limitations of the research results, and puts forward the ideas for the next step. It is suggested to add application tests in other related fields and predict the future development and application.
Reviewer 2 Report
Authors proposed novel Graph based deep neural network for unstructured data. Work seems interesting and original yet some of the issues should be corrected before publication.
1. Split the introduction section into two: Introduction and Related work. In related work you should describe already published articles from litrerature on Graph convolutional neural networks. Your related work is focused on your method and thus should be in Method section.
2. Make better headings system. use subsections eg. 1.1, 1.2., 2.1 and so on.
3. Images should be in better format.
4. Font of equations should be in better format.
5. More state of art references should be used.
6. What are the CNN specifications? Can you describe your CNN model used for experiments? Number of layers number of parameters?
7. Can you provide some comparison between number of operations (parameters) for different models used in experiments?
8. Can you provide some more insights into future work?
Round 2
Reviewer 2 Report
The authors addressed all issues raised accordingly. I can now suggest publishing of the paper.